# Crude Glycerol/Guishe Based Catalysts for Biodiesel Production: Conforming a Guishe Biorefinery

**Andrea Belén Figueroa-Díaz, Salvador Carlos-Hernández and Lourdes Díaz-Jiménez ***

Laboratorio de Revaloración de Residuos, Sustentabilidad de los Recursos Naturales y Energía, Cinvestav Saltillo, Ramos Arizpe, Coahuila 25900, Mexico; belen.figueroa@cinvestav.edu.mx (A.B.F.-D.); salvador.carlos@cinvestav.edu.mx (S.C.-H.)
* Correspondence: lourdes.diaz@cinvestav.edu.mx; Tel.: +52-844-438-9600

**Abstract:** Biodiesel production imposes some challenges, such as the crude glycerol management and cleaning requirements of biodiesel produced by homogeneous transesterification. Heterogeneous catalysts based on residual biomass have been proposed to tackle these challenges; in addition, biomass revalorization is fundamental for biorefineries development. In this research, two organic wastes (crude glycerol and guise) are used to synthesize carbonaceous catalysts. Four catalysts, with different crude glycerol/guishe proportions, were prepared by pyrolysis at 800 and 900 °C, followed by a chemical functionalization with $H_2SO_4$. Scanning electron microscopy (SEM), Fourier transform infrared spectroscopy (FT-IR), and thermal gravimetric analysis (TGA) were used to characterize the catalysts. The performance of the catalysts was evaluated in a soybean oil transesterification reaction. The crude glycerol/guishe based catalysts lead to similar biodiesel yields than the obtained with a conventional homogeneous catalyst ($CH_3NaO$). The catalyst identified as BS-25-8 (a mixture of 25% guishe and 75% crude glycerol, pyrolyzed at 800 °C and sulfonated), in a proportion of 1 wt%, achieved the highest fatty acid methyl esters (FAME) yield (99%) in the transesterification reaction, even surpassing the performance of the $CH_3NaO$ (yield of 93%).

**Keywords:** biochar; sulfonation; FAME; transesterification; heterogeneous catalysis

## 1. Introduction

Due to growing social development, fossil fuel consumption to supply energy needs is continuously increasing. This situation has generated environmental issues such as climate change, fossil fuel depletion, and increasing $CO_2$ and other greenhouse gases (GHGs) emissions [1,2]. Therefore, renewable energies, especially biofuels, play a crucial role in reducing fossil fuel dependence and the associated environmental impacts. Among biofuels, biodiesel has been an alternative to reduce the use of conventional diesel (CD). Biodiesel is a fuel approved by the Environmental Protection Agency. It has been shown that by using biodiesel instead of CD, the net $CO_2$ emissions are reduced up to 78% [3]. Two types of reactions can produce biodiesel: (1) free fatty acids (FFA) esterification and (2) vegetable oils (VO) or residual oils (RO) transesterification. Transesterification, the standard method of biodiesel production at an industrial level, involves oil reaction with low molecular weight alcohol (methanol or ethanol). This reaction is promoted by a catalyst (acid or alkali), generating fatty acid methyl/ethyl esters (FAME/FAEE) as a product and crude glycerol as a byproduct [3,4].

Unfortunately, some issues arise during the production process of this biofuel. For example, the commonly used homogeneous alkali catalysts (sodium hydroxide or potassium hydroxide) are limited to reacting with virgin vegetable oils (VO) since using residual oils (RO) leads to the generation of soaps. These are mixed with the product, which requires extra steps for its separation involving high volumes of water [5]. Similarly, although homogeneous acid catalysts can be used with RO, they are highly corrosive to reactors [6]. Another issue is the continuous generation of crude glycerol (CG) as a byproduct derived

from the transesterification reaction. Crude glycerol has limited applications due to its impurities, such as residues of the homogeneous catalyst used in the reaction [7–9].

Based on the above facts, the integral use of biomass resources, through transformation platforms currently known as biorefineries, is an alternative to solving biodiesel production problems. A characteristic of such platforms is the production of biofuels and bioproducts.

Recently, biochar synthesis from residual biomass has been studied with application in transesterification reactions, showing advantages over homogeneous catalysts, such as low cost, lower catalyst consumption, and the ability to perform simultaneous reactions [5].

A heterogeneous transesterification reaction follows a sequential mechanism that involves seven steps: (1) mass transfer of reagents from the fluid to the external surface of the catalyst, (2) diffusion of the reagents through the porosity of the catalyst, (3) adsorption of the reagents on the surface of the catalyst, (4) reaction on the catalyst surface, (5) desorption of surface products, (6) diffusion of the products from the interior of the catalyst to the external surface, and (7) mass transfer of the products from the external surface of the particle to the fluid [10]. Since the reaction occurs on the catalyst surface, it is important to develop functional groups on the surface to improve the catalyst activity [11,12].

In this sense, a series of heterogeneous catalysts were obtained from bovine bone, crab shell, and eggshell. These catalysts were assayed in transesterification reaction achieving FAME yields between 94 and 97% [13]. In other studies, the functionalization of alkali carbonaceous catalysts by acid activation has been proposed to achieve simultaneous reactions [5]. González et al. obtained ~90% FAME yields using a synthesized catalyst from pyrolyzed oat hull functionalized with sulfuric acid [14].

The availability of raw materials for the synthesis of heterogeneous catalysts should be analyzed. Crude glycerol is a residue derived from the VO transesterification reaction, representing technical and economic issues for the biodiesel industry [9]. It is composed of ~ 65–85 wt% of glycerol, and the rest represents impurities such as inorganic salts, non-glycerol organic matter, and water [15]. On the other hand, in the obtaining of ixtle (natural fiber from *Agave lechuguilla*), a waste called "guishe" (GI) is produced, which induces environmental issues [16]. Ixtle represents 14.6% of the plant, while GI corresponds to 85.4% [17]. The GI is mainly composed of cellulose (17.72% dry basis), hemicellulose (17.15% dry basis), and lignin (7.32% dry basis) [18], which is suitable for obtaining biochar by thermochemical treatment. In previous work, a guishe biorefinery was conceptualized as a strategy to promote the integral revalorization of this byproduct; among the bioproducts which could be obtained from guishe transformation, there are saponins, phytochemicals, ash, and biochars [16]. Ash and biochars could be used in different applications such as adsorption, soil remediation, and catalysis.

Then, to follow up on that biorefinery strategy, the objective of this study was to develop a catalyst from the mixture of two wastes: crude glycerol (CG) from biodiesel production and lignocellulose biomass "guishe" (GI). The performance of the developed catalyst was assayed on the soybean oil transesterification reaction.

## 2. Results and Discussion

### 2.1. Elemental Analysis of Raw Materials

Table 1 includes the results of the proximal analysis and the chemical composition of both raw materials. The moisture value found in CG is 1.74%, which agrees with the reported in other works; this percentage is due to CG composition, which contains glycerol, alcohol, impurities, and a low concentration of water [15,19]. The ash proportion is larger than the reported by [19]. This parameter depends on the oil used in transesterification. The CG used in this study has many impurities since it is obtained as a byproduct from a biodiesel production process when residual oils were the raw material. In contrast, fixed carbon is also larger than reported by Pachapur due to the origin of the crude glycerol [19]. Regarding GI, the moisture content is similar to the obtained in other works [18]. This low value is adequate since it implies that the raw material requires

low energy to be transformed by thermal processes. In contrast, the fixed carbon is low compared to the reported for agriculture wastes [18].

**Table 1.** Proximal and elemental analysis of CG and GI.

|  |  | Crude Glycerol | Guishe |
|---|---|---|---|
| Proximal analysis (%) | Moisture | 1.74 | 4.15 |
|  | Volatiles | 17.25 | 15.18 |
|  | Ash | 6.25 | 32.15 |
|  | Fixed carbon | 74.73 | 48.48 |
| Elemental analysis (%) | Ca | <0.001 | 28.6 |
|  | K | 3.13 | 0.62 |
|  | Mg | <0.001 | 0.39 |
|  | Si | nd | 0.997 |
|  | Fe | nd | 0.45 |
|  | Al | nd | 0.41 |
|  | S | nd | 0.28 |
|  | P | nd | 0.27 |

Calcium was found in the highest concentration (28.56%) in GI. This amount is directly proportional to the calcium oxalate content reported in *A. lechuguilla* plant [20]. It has been reported that a calcium rich catalyst shows a good performance in biodiesel production using VO due to the alkalinity required to carry out transesterification reactions [21,22]. Then, the presence of this element in guishe represents an advantage for this application. On the other hand, potassium was the highest concentration (3.13%) element in CG; this is due to the KOH residues used as a homogeneous catalyst to produce biodiesel (Table 1).

### 2.2. Pyrolysis Conditions Determination

Figure 1 shows the TGA and differential thermal analysis (DTA) of CG, GI (M-100) and mixtures of them (M-75 GI:CG = 75:25; M-50 GI:CG = 50:50; M-25 GI:CG = 25:75). Four decomposition phases are observed in all materials. In phase 1, the most significant weight loss (4%) occurred in M-100 between 30–170 °C, which corresponds to water evaporation and volatile matter degradation [23]. For M-50 and M-75, this weight loss was lower (2%), where in addition to evaporating water, methanol decomposition was present since the CG content [24]. In phase 2, between 150 and 350 °C, similar behavior is observed; CG showed the highest weight loss (~50%); according to Dou et al. [24], these results can be attributed to the degradation of glycerol in CG [24] and hemicellulose for GI [18].

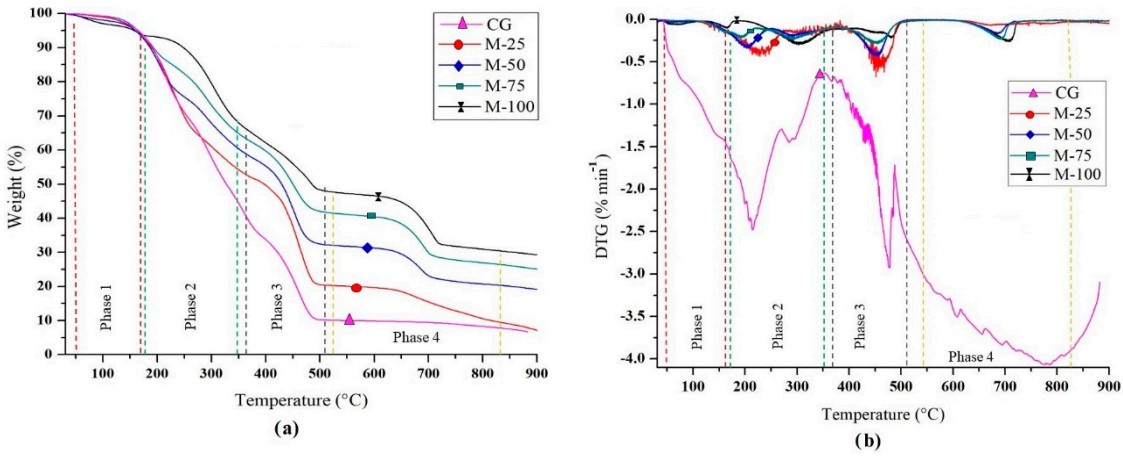

**Figure 1.** (**a**) Thermogravimetric analysis, and (**b**) differential thermogravimetric analysis of M-100, M-75, M-50, M-25, and CG.

Between 280 and 550 °C (Phase 3), CG presented a weight loss of ~30% due to the degradation of fatty acids, methyl esters, and tar. Finally, in phase 4, the mixtures and M-100 exhibited a mass loss associated with lignin degradation.

It is observed that from ~750 °C the degradation of evaluated materials was negligible, so it was decided to perform the pyrolysis process at 800 and 900 °C.

Figure 2 shows that biochar yields decreased with increasing temperature pyrolysis. According to Xie et al. [25], in the biomass pyrolysis process, labile organic matter is degraded at low temperatures. In the case of this study, the labile matter corresponds to hemicellulose, volatiles, glycerol, etc. Moreover, mineralized fixed carbon (such as cellulose in GI and fatty acids in CG) is hardly affected because the temperatures required to transform more stable forms of C should be higher. In addition, as the temperature increases, other components (lignin from guishe and impurities of CG) produce the final fixed organic carbon. Therefore, the lower yield of biochars obtained at 900 °C can be attributed to a more significant loss of mineralized fixed carbon, in addition to the destruction of labile C. The energy-dispersive X-ray spectroscopy (EDS) results corroborate this since the C content decreases in the biochar treated at 900 °C. In this sense, an inverse relationship between the temperature and biochar yield was found, and a higher yield was obtained by using a lower temperature.

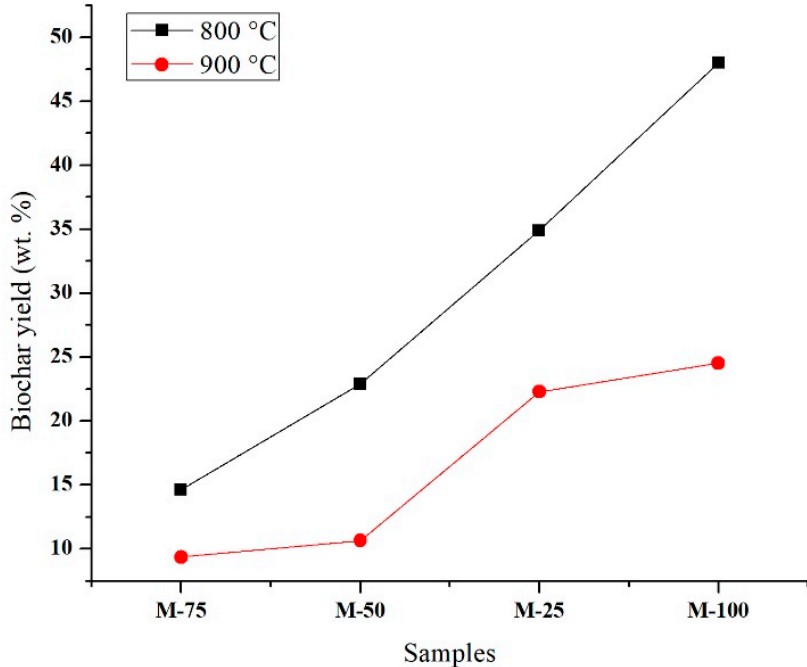

**Figure 2.** Effect of temperature on samples M-100, M-25, M-50, M-75.

### 2.3. Composition and Morphology by Microscopy

Catalysts obtained at 800 °C: B-100-8 (Figure 3a) shows pore formation generated by the thermal degradation of GI fibers. Also, the presence of calcium oxalate particles is visualized, which agrees with the EDS results (Table 2). This biochar had the highest percentage of calcium (22.17%) compared to all biochars obtained at 800 °C; this can be attributed to calcium in the soil where *A. lechuguilla* grows. The effect of sulfuric acid treatment on the surface of BS-100-8 is shown in Figure 3b. The formation of ordered pores in a honeycomb arrangement with pore diameter ~2.56 μm is remarkable. It has been reported that the acid treatment promotes the solubilization of hemicellulose keeping intact the lignin and must of the cellulose [26]. Therefore, functionalization changed the material pore size, as mentioned by Xiong et al. [27]. In B-25-8 (Figure 3c), the presence of the fiber is no longer appreciated while the persistence of agglomerates stands out; its porosity is affected by the excess of CG. However, sulfonation modifies the morphology

by increasing its porosity with pore size ~2.82 µm, as shown in Figure 3d. The porosity of B-50-8 is very different from the others (Figure 3e); this can be attributed to the content of volatile material in CG since the pores took the form of gas bubbles that tried to escape during the thermochemical process. In the case of BS-50-8, when functionalization occurred (Figure 3f), its porosity is lower than that of all sulfonated biochars (≈597 nm). Finally, in B-75-8 (Figure 3g), the porosity of the fiber becomes more pronounced compared to B-100-8. This could be attributed to the expansion of GI fibers due to the quick thermal decomposition of CG glycerol. When sulfonation occurred, the porosity of BS-75-8 changes completely compared with B-75-8, channels with dimensions of ≈4.40 µm can be observed (Figure 3h).

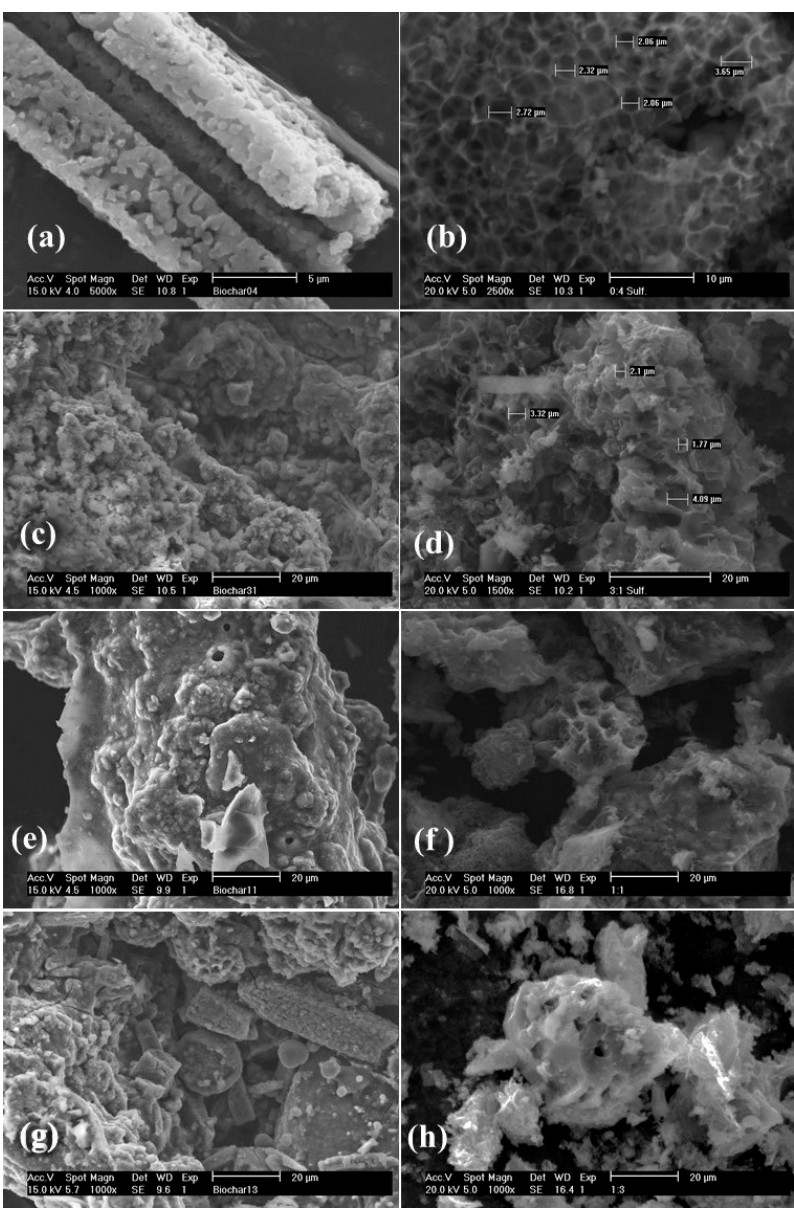

**Figure 3.** Micrographs of catalysts obtained at 800 °C: (**a**) B-100-8, (**b**) BS-100-8, (**c**) B-25-8, (**d**) BS-25-8, (**e**) B-50-8, (**f**) BS-50-8, (**g**) B-75-8, (**h**) BS-75-8.

**Table 2.** Elemental composition of the biochars obtained at 800 °C.

| Catalysts | | Elements (%) | | | | | | | | |
|---|---|---|---|---|---|---|---|---|---|---|
| | | **C** | **O** | **Mg** | **Al** | **Si** | **P** | **S** | **K** | **Ca** |
| Biochars | B-25-8 | 45.6 | 18.0 | 0.5 | 0.1 | 0.3 | 0.2 | 0.2 | 19.1 | 15.0 |
| | B-50-8 | 41.8 | 17.7 | 0.9 | 0.3 | 1.4 | 0.2 | 0.1 | 15.5 | 21.7 |
| | B-75-8 | 50.6 | 17.5 | 0.6 | 0.1 | 0.5 | 0.3 | 0.2 | 4.7 | 24.6 |
| | B-100-8 | 44.5 | 19.6 | 1 | 0.5 | 1.4 | 0.4 | 0.2 | 4.2 | 22.1 |
| Sulfonated biochars | BS-25-8 | 55.1 | 1.3 | 2.4 | 2.8 | 8.4 | 2.2 | 14.7 | 5.6 | 6.6 |
| | BS-50-8 | 57.0 | 1.7 | 2.2 | 3.5 | 5.1 | - | 19.1 | 2.5 | 8.1 |
| | BS-75-8 | 76.3 | 1.1 | - | 4.8 | - | - | 8.7 | 2.8 | 5.6 |
| | BS-100-8 | 64.4 | 1.9 | 3.9 | 3.3 | 9.7 | 9.2 | 9.2 | 2.9 | 14.6 |

When carrying out the elemental analysis of the sulfonated biochars obtained at 800 °C, an increase of sulfur content was observed (Table 2). This increment could be associated with the formation of -SO$_3$H groups on the biochar surface [14].

Catalysts obtained at 900 °C: In catalyst B-100-9 (Figure 4a), the degradation is more perceptible than the observed in the material treated at 800 °C (B-100-8). A decrease in the volume of the fibers, a different porosity, and the presence of pronounced fractures were observed.

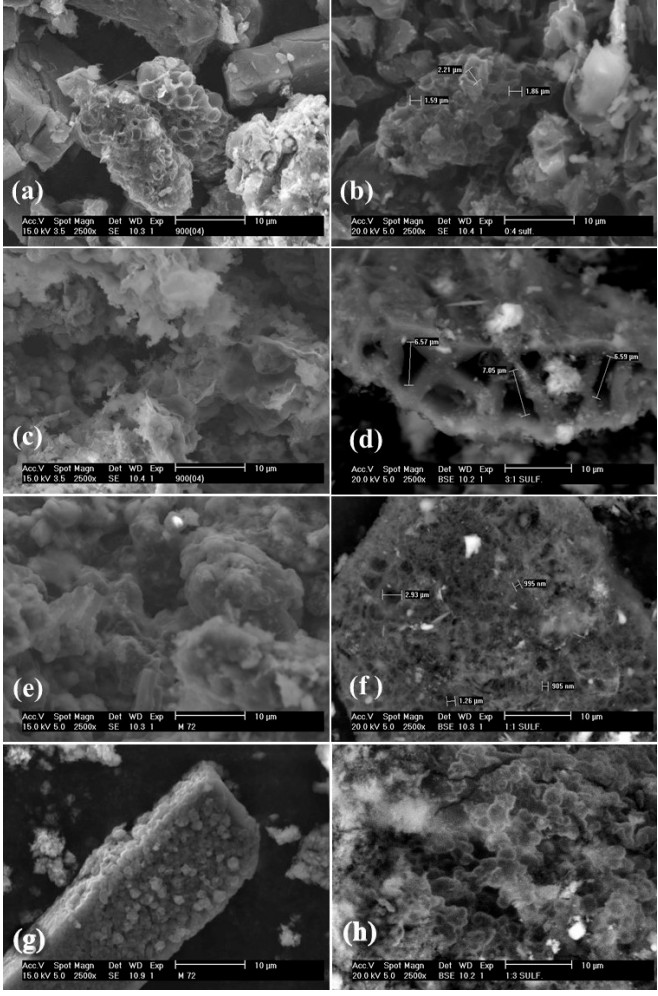

**Figure 4.** Micrographs of biochars obtained at 900 °C: (**a**) B-100-9, (**b**) BS-100-9, (**c**) B-25-9, (**d**) BS-25-9, (**e**) B-50-9, (**f**) BS-50-9, (**g**) B-75-9, (**h**) BS-75-9.

On the other hand, the calcium content in B-100-9 is higher than B-100-8 (Table 3); this can be attributed to calcium oxide formation. Moreover, Figure 4b shows a change in the porosity of BS-100-9, even in calcium oxide particles induced by sulfonation.

**Table 3.** Elemental composition of the biochars obtained at 900 °C.

| Biochar | | Elements (%) | | | | | | | | |
|---|---|---|---|---|---|---|---|---|---|---|
| | | C | O | Mg | Al | Si | P | S | K | Ca |
| Biochars | B-25-9 | 25.5 | 22.8 | 1.4 | 0.8 | 2.9 | 0.5 | 0.2 | 17.9 | 27.4 |
| | B-50-9 | 37.1 | 17.7 | 0.9 | 0.3 | 1.4 | 0.3 | 0.1 | 15.5 | 21.7 |
| | B-75-9 | 43.5 | 18.0 | 0.8 | 0.6 | 2.6 | 0.3 | 0.1 | 6.3 | 26.6 |
| | B-100-9 | 39.1 | 20.0 | 1.2 | 0. | 0.9 | 0.6 | 0.2 | 3.8 | 33.5 |
| Sulfonated biochars | BS-25-9 | 58.3 | 17.3 | - | - | 2.0 | - | 7.7 | 1.1 | 13.2 |
| | BS-50-9 | 73.1 | 14.3 | - | - | 0.3 | - | 3.8 | 0.8 | 7.4 |
| | BS-75-9 | 83.1 | 8.1 | - | - | 0.3 | - | 3.3 | 0.4 | 4.6 |
| | BS-100-9 | 65.8 | 3.4 | 0.3 | 1.5 | 11.2 | - | 7.3 | 2.4 | 4.7 |

The degradation behavior is similar in catalysts (B-25-9, BS-25-9, B-50-9, BS-50-9, B-75-8, and BS-75-8). The sulfur content is less the half in sulfonated catalysts obtained at 900 °C than the catalysts treated at 800 °C (Table 3). This is probably due to the effect of the temperature; at 900 °C sulfur compounds are released.

### 2.4. Fourier Transform Infrared Spectroscopy

Figure 5 shows the FT-IR spectra of the sulfonated biochars. The signal centered at 2351 cm$^{-1}$ in Figure 5a is associated with the C-H bonds stretching, which correspond either to the hydrocarbons CG or the alkyl chains in GI cellulose [28]. On the other hand, the signal centered at 3128 cm$^{-1}$ corresponds to -OH stretches of the carboxyl and phenolic hydroxyl groups. The signal at 1621 cm$^{-1}$ suggests the formation of acid groups due to sulfonation [14,28].

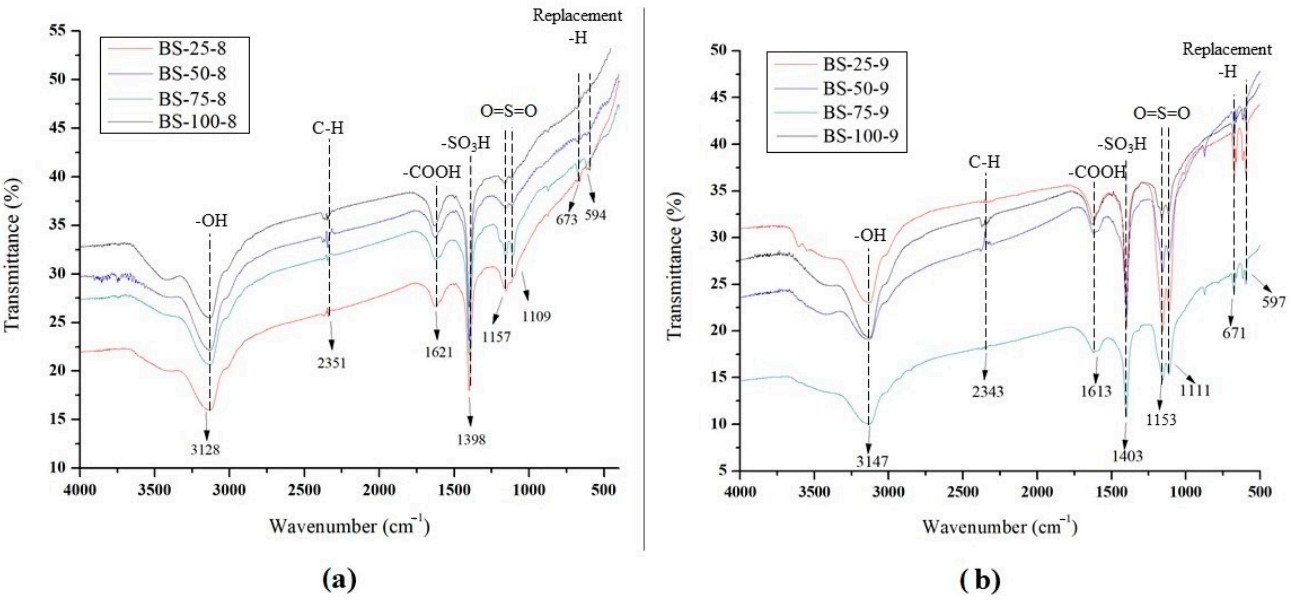

**Figure 5.** FT-IR of the functionalized biochars obtained at (**a**) 800 °C and (**b**) 900 °C.

In this sense, the signal at 1398 cm$^{-1}$ is likely due to the superposition of -COOH and stretching of -SO$_3$H, according to different reports [29,30]. Sulfonation is corroborated by the presence of the the signal at 1109 cm$^{-1}$, which is associated with O=S=O symmetric and SO$_3$$^-$ stretching modes of –SO$_3$H group, while the peak at 1157 cm$^{-1}$ attributed to SO$_2$ stretching indicates the presence of SO$_3$H [31].

Finally, the signals observed at 594 and 673 cm$^{-1}$ could imply replacing the -H group on the aromatic ring by -SO$_3$H. In this way, the formation of the sulfonic functional groups on the surface of the biochar was confirmed [28].

On the other hand, the biochars spectra obtained at 900 °C are presented in Figure 5b. The behavior of the FT-IR spectra of these catalytic biochars is like the obtained at 800 °C, with the difference that the signal corresponding to -SO$_3$H stretching is less pronounced, which agrees with the observed by SEM. However, the signals at the SO$_2$ stretching in biochars obtained at 900 °C are more intense. A possible explanation of this behavior is as follows. It is assumed that both the CG/GI ratio and temperature affect the degree of graphitization. Since the mixed materials are exposed to heat in an inert atmosphere, compounds such as the volatiles and even the impurities of glycerol could react with the calcium of guishe, leading to different stable structures of the biochars. Then, these structures are affected in different ways by the sulfuric acid producing at once different distributions of sulfonic groups. However, a deeper analysis of the graphitization degree is required to elucidate this mechanism.

### 2.5. Transesterification Assays

#### 2.5.1. Biodiesel Yield

Using the reference catalyst (CH$_3$NaO) in 1 wt%, an ~85% yield was reached. The reaction yield increases by up to 93% when the amount of catalyst is augmented to 3 wt%.

On the other hand, using 1 wt% of catalytic biochars obtained at 800 °C (Figure 6a), the biodiesel yield is similar to that obtained with the reference catalyst. The higher yield corresponds to the BS-50-8 catalyst (93%), followed by BS-75-8 (92%) and BS-25-8 (79%). Therefore, these catalytic biochars represent an alternative for the revalorization of crude glycerol and guishe. When applying 3 wt% heterogeneous catalysts, the BS-25-8 biochar gave the highest yield (99%) of all cases.

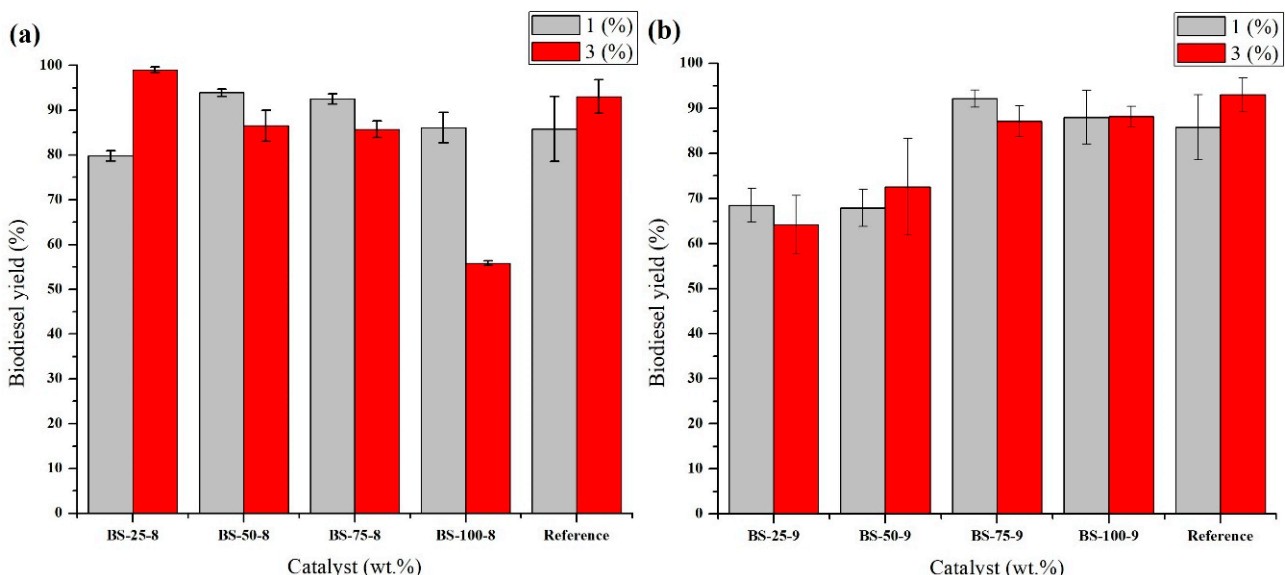

**Figure 6.** Biodiesel yield (%) using the catalysts obtained at (**a**) 800 °C, and (**b**) 900 °C.

This inverse relationship between conversion and catalyst load could be due to the potentiated activity of the catalysts. That means 1% of the catalyst is enough to achieve the conversion of triglycerides, but 3% already implies an excess of reactivity leading to the breakdown of the fatty acid methyl esters to alkyl chains.

The results obtained by the catalytic biochars synthesized at 900 °C are shown in Figure 6b. When using 1 wt% catalyst, the biochar BS-100-9 (88%) and BS-75-9 (92%) surpass the homogeneous catalyst performance, while the biochars BS-25-9 and BS-50-9 are below the reference yield with yields of 68 and 67%, respectively. Using a 3 wt% catalyst, none of the biochars synthesized at 900 °C achieve the yield obtained with $CH_3NaO$, although the biochars BS-100-9 and BS-75-9 are close (88% and 87%, respectively).

The differences in the biodiesel yield are induced by both the CG/GI ration and pyrolysis temperature; for 900 °C, more guishe in the biochar leads to higher biodiesel yield considering a catalyst load of 3%. A similar trend is observed when the catalyst load is 1%, but the maximum biodiesel yield is obtained for a 25/75 ratio. Further, for 800 °C, more guishe in the biochar leads to lower biodiesel yield considering a catalyst load of 3% (inverse than the obtained for 900 °C). When the catalyst load is 1%, maximum yields are achieved by using a higher concentration of guishe in the mixture. From this comparison, it is deduced that the biochars pyrolyzed at 800 °C allow obtaining higher biodiesel yields.

### 2.5.2. FAME Analysis

To evaluate the influence of catalysts on biodiesel quality, the analysis by gas chromatography was carried out.

For comparative purposes, the biodiesel obtained using $CH_3NaO$ was also analyzed. Around 20 compounds were identified. Most of them correspond to FAME (biodiesel). Table 4 shows the four major compounds identified in the biodiesel produced by the homogeneous catalyst. The contribution of these four compounds reaches 79.9 and 91.4% when using 1 and 3 wt% of catalyst, respectively. Of these compounds, methyl linoleate was the main component, which agrees with findings reported by Chellappan et al. [32].

**Table 4.** Results of the GC–MS analysis of the biodiesel produced using the homogeneous catalyst $CH_3NaO$ at 1 and 3 wt%.

| Catalyst | Cat (% wt) | $t_{ret}$ (min) | Methyl Ester | Condensed Formula | *m/z* | Area (%) |
|---|---|---|---|---|---|---|
| $CH_3NaO$ | 1 | 10.33 | Methyl palmitate | $C_{17}H_{34}O_2$ | 270 | 9.12 |
| | | 12.77 | Methyl linoleate | $C_{19}H_{34}O_2$ | 294 | 45.39 |
| | | 12.90 | Methyl oleate | $C_{19}H_{36}O_2$ | 296 | 20.96 |
| | | 13.33 | Methyl stearate | $C_{19}H_{38}O_2$ | 298 | 4.43 |
| | 3 | 10.34 | Methyl linoleate | $C_{17}H_{34}O_2$ | 270 | 9.50 |
| | | 12.83 | Methyl oleate | $C_{19}H_{34}O_2$ | 294 | 54.62 |
| | | 12.95 | Methyl stearate | $C_{19}H_{36}O_2$ | 296 | 21.28 |
| | | 13.37 | Methyl linoleate | $C_{19}H_{38}O_2$ | 298 | 5.99 |

Regarding the sulfonated catalysts obtained at 800 °C (Table 5), the BS-100-8 and BS-25-8 catalysts show four and seven types of fatty acid methyl esters, respectively. This result represents an advantage over the $CH_3NaO$ catalyst (20 compounds); it could indicate that the heterogeneous transesterification reaction focused on transforming the triglycerides of the oil to FAME without reaching the breakdown of the alkyl chains. The methyl ester of 9,12-octadecadienoic acid was the primary compound. It is essential to mention that when using 1 wt% of the BS-100-8 catalyst, the four identified compounds represent 100% of FAME, while with 3 wt%, the four compounds reach only 70% of FAME. Therefore, it can be deduced that using biochars is possible to produce better quality biodiesel than the homogeneous catalyst.

**Table 5.** GC–MS analysis of the biodiesel produced using the catalysts BS-100-8, BS-25-8, BS-50-8 and BS-75-8 at 1 and 3 wt%.

| Catalyst | Cat (% wt) | Methyl Ester | Condensed Formula | Area (%) |
|---|---|---|---|---|
| BS-100-8 | 1 | Methyl palmitate | $C_{17}H_{34}O_2$ | 17.47 |
| | | Methyl linoleate | $C_{19}H_{34}O_2$ | 50.21 |
| | | Methyl oleate | $C_{19}H_{36}O_2$ | 27.48 |
| | | Methyl stearate | $C_{19}H_{38}O_2$ | 4.84 |
| | 3 | Methyl palmitate | $C_{17}H_{34}O_2$ | 9.65 |
| | | Methyl linoleate | $C_{19}H_{34}O_2$ | 33.57 |
| | | Methyl oleate | $C_{19}H_{36}O_2$ | 18.11 |
| BS-25-8 | 1 | Methyl palmitate | $C_{17}H_{34}O_2$ | 6.26 |
| | | Methyl linoleate | $C_{19}H_{34}O_2$ | 45.88 |
| | | Methyl stearate | $C_{19}H_{38}O_2$ | 3.35 |
| | 3 | Methyl pentadecanoate | $C_{17}H_{34}O_2$ | 12.48 |
| | | Methyl linoleate | $C_{19}H_{34}O_2$ | 45.09 |
| | | Methyl oleate | $C_{19}H_{36}O_2$ | 22.92 |
| | | Methyl stearate | $C_{19}H_{38}O_2$ | 3.70 |
| BS-50-8 | 1 | Methyl palmitate | $C_{17}H_{34}O_2$ | 7.66 |
| | | Methyl linoleate | $C_{19}H_{34}O_2$ | 41.13 |
| | 3 | Methyl palmitate | $C_{17}H_{34}O_2$ | 21.03 |
| | | Methyl linoleate | $C_{19}H_{34}O_2$ | 17.13 |
| | | Methyl petroselinate | $C_{19}H_{36}O_2$ | 13.44 |
| BS-75-8 | 1 | Methyl linoleate | $C_{19}H_{34}O_2$ | 24.74 |
| | | Methyl octadecenoate | $C_{19}H_{36}O_2$ | 12.40 |
| | 3 | Methyl palmitate | $C_{17}H_{34}O_2$ | 6.33 |
| | | Methyl linoleate | $C_{19}H_{34}O_2$ | 16.19 |
| | | Methyl octadecenoate | $C_{19}H_{36}O_2$ | 12.25 |

On the other hand, the BS-25-8 catalyst obtained five and eight peaks using 1 and 3 wt% catalysts, respectively. Among the four major compounds, the maximum percentage by mass corresponds to the methyl ester of acid 9, 12-octadecadienoic. The four major compounds represent 61% (1 wt%) and 84% (3 wt%) of FAME.

On the other hand, Table 6 shows the chromatography results when using the sulfonated catalysts obtained at 900 °C. The catalysts BS-100-9 and BS-25-9 show 7 and 18 FAME, respectively. Comparing the results obtained with the catalysts produced at 800 °C, the number of peaks increased, observing FAME's lower production. However, among the four major compounds, methyl linoleate (9,12-Octadecadienoic acid, methyl ester) was the one with the highest concentration in the two percentages (1 and 3 wt%). This is because the primary fatty acid present in soybean oil is linoleic acid.

The majority compound in almost all the cases was methyl linoleate, obtaining the highest percentage (50%) with the BS-100-8 catalyst at 1 wt% compared to the homogeneous catalyst (45%). Besides, the BS-25-8 catalyst matched the reference catalyst to 1 wt% with a mass percent (45%).

**Table 6.** GC–MS analysis of the biodiesel produced using the catalysts BS-100-9, BS-25-9, BS-50-9 and BS-75-9 at 1 and 3 wt%.

| Catalyst | Cat (% wt) | Methyl Ester | Condensed Formula | Area (%) |
|---|---|---|---|---|
| BS-100-9 | 1 | Cyclopentanol undecanoic acid, methyl ester | $C_{17}H_{34}O_2$ | 10.66 |
| | | Butyl octyl phthalate | $C_{20}H_{30}O_4$ | 21.96 |
| | | Methyl linoleate | $C_{19}H_{34}O_2$ | 23.76 |
| | | Methyl octadecanoate | $C_{19}H_{36}O_2$ | 15.47 |
| | 3 | Butyl octyl phthalate | $C_{20}H_{30}O_4$ | 35.21 |
| | | Methyl linoleate | $C_{19}H_{34}O_2$ | 19.06 |
| | | Methyl petroselinate | $C_{19}H_{36}O_2$ | 13.87 |
| | | Tert-butyl palmitate | $C_{20}H_{40}O_2$ | 12.06 |
| BS-25-9 | 1 | Methyl palmitate | $C_{17}H_{34}O_2$ | 7.84 |
| | | Butyl octyl phthalate | $C_{20}H_{30}O_4$ | 12.17 |
| | | Methyl linoleate | $C_{19}H_{34}O_2$ | 20.38 |
| | | Methyl octadecanoate | $C_{19}H_{36}O_2$ | 13.99 |
| | 3 | Methyl palmitate | $C_{17}H_{34}O_2$ | 8.67 |
| | | Butyl octyl phthalate | $C_{20}H_{30}O_4$ | 8.55 |
| | | Methyl linoleate | $C_{19}H_{34}O_2$ | 34.47 |
| | | Methyl petroselinate | $C_{19}H_{36}O_2$ | 19.11 |
| BS-50-9 | 1 | Methyl palmitate | $C_{17}H_{34}O_2$ | 5.67 |
| | | Butyl octyl phthalate | $C_{20}H_{30}O_4$ | 10.44 |
| | | Methyl linoleate | $C_{19}H_{34}O_2$ | 14.43 |
| | | Methyl octadecanoate | $C_{19}H_{36}O_2$ | 12.92 |
| | 3 | Methyl palmitate | $C_{20}H_{30}O_4$ | 9.66 |
| | | Butyl octyl phthalate | $C_{19}H_{34}O_2$ | 16.06 |
| | | Methyl linoleate | $C_{19}H_{36}O_2$ | 29.19 |
| | | Methyl octadecanoate | $C_{20}H_{40}O_2$ | 22.62 |
| BS-75-9 | 1 | Methyl palmitate | $C_{17}H_{34}O_2$ | 6.37 |
| | | Butyl octyl phthalate | $C_{20}H_{30}O_4$ | 21.12 |
| | | Methyl linoleate | $C_{19}H_{34}O_2$ | 30.75 |
| | | Methyl petroselinate | $C_{19}H_{36}O_2$ | 9.37 |
| | 3 | Cyclopentanol undecanoic acid, methyl ester | $C_{17}H_{34}O_2$ | 7.06 |
| | | Methyl linoleate | $C_{20}H_{30}O_4$ | 19.75 |
| | | Methyl linoleate | $C_{19}H_{34}O_2$ | 19.17 |
| | | Terc-butil palmitate | $C_{19}H_{36}O_2$ | 10.13 |

## 3. Materials and Methods

### 3.1. Materials and Treatments

The CG was provided by the Mexican company Enremex (Tijuana, Mexico), and it was used without pre-treatment. The GI was collected from the ixtle production area of Ramos Arizpe, Coahuila, Mexico; this was previously dried in an oven at 60 °C for 48 h and ground to powder in a blade mill. After that, it was sieved with mesh No. 60.

Three mixtures with different mass proportions (75:25, 50:50, 25:75) of the CG and GI materials were prepared and identified as M-25, M-50, and M-75, respectively. Also, for comparison, additional material was considered. This consisted of just GI, which was identified as M-100.

The procedure was as follows; the sieved GI was mixed with CG previously heated at 30 °C for 15 min to decrease its viscosity.

Proximal analysis of raw material was performed following the standards ISO 18134-2:2015, ISO 18123:2015, and ISO 18122:2015 for moisture, volatiles, and ashes, respectively.

### 3.2. Biochar Synthesis

M-25, M-50, M-75, and M-100 were treated by slow pyrolysis using a horizontal tubular furnace at 800 and 900 °C for 1 h in $N_2$ (INFRA; Saltillo, Mexico) atmosphere. Catalysts obtained at 800°C were identified as B-25-8, B-50-8, B-75-8, and B-100-8, while the materials obtained at 900 °C were identified as B-25-9, B-50-9, B-75-9, and B-100-9.

### 3.3. Biochar Functionalization

The sulfonation stage was performed adapting the treatment described in reported works [28,33,34] using sulfuric acid (97.7%) (Sigma-Aldrich; St Louis, MO, USA).

The biochars were placed in a beaker, and sulfuric acid was added; the mixtures were stirred for 15 min at room temperature. After that, the treated biochars were recovered by decantation and washed three times with distilled water. The remaining solids were placed in crucibles and heated at 100 °C for 15 h, consecutively; they were washed with distilled water up to neutral pH. After that, the catalysts were dried at 110 °C for 12 h. Sulfonated biochars obtained at 800 °C were named as BS-25-8, BS-50-8, BS-75-8, and BS-100-8; and BS-25-9, BS-50-9, BS-75-9, and BS-100-9 when they were thermally treated at 900 °C.

### 3.4. Transesterification Reaction

Transesterification reaction was carried out following a methodology previously reported [35]. It was performed at $65 \pm 2$ °C for 4 h at 800 rpm, with a molar ratio of 1:9 (oil/methanol), considering 1 and 3 wt% of catalyst. Sodium methoxide ($CH_3NaO$) (Sigma-Aldrich; St Louis, MO, USA) was considered as a homogeneous reference catalyst. The catalyst was loaded in a flat bottom flask, then methanol and oil were added. Flasks were placed in a water bath at 70 °C under magnetic stirring for 4 h. After the reaction time has elapsed, the samples were let stand for 24 h for the separation of biodiesel/glycerol (phases). In reactions using catalytic biochar, it was necessary to use centrifugation (model C600A; SOLBAT, Puebla, Mexico) to phase separation. The phase separation was carried out by decantation. Subsequently, they were subjected to drying in an oven at 65 °C for 24 h in order to eliminate the residual methanol. Subsequently, the volume of the phases obtained was measured to determine the biodiesel yield.

### 3.5. Experimental Design

A full factorial design ($2 \times 2 \times 4$) was applied, and the assays were performed in triplicates. The investigated factors were A: % of catalyst (two levels), B: pyrolysis temperature (two levels), C: catalyst (with four levels) as a reference catalyst was considered sodium methoxide ($CH_3NaO$).

### 3.6. Characterization

K, Ca, and Mg contents of CG were determined using plasma emission spectrophotometry (ICP OPTIMA 8300, PerkinElmer; Waltham, MA, USA). In the case of GI, semiquantitative X-ray fluorescence analyses were performed (Bruker S4 Pionner, Billerica, MA, USA).

Thermogravimetric analyses were performed from 30 to 900 °C at 7 °C $min^{-1}$, under $N_2$ atmosphere (SDT-Q600, TA Instruments, New Castle, DE, USA). The technique was applied to choose the pyrolysis conditions. A comparison was also performed between the biochar yield obtained at 800 and 900 °C.

The elemental surface composition, along with morphology, was analyzed by scanning electron microscopy combined with energy dispersion spectroscopy (XL-30 ESEM, Philips; North Billerica, MA, USA). Fourier Transform Infrared Spectroscopy, at a resolution of 4 $cm^{-1}$, was performed on CG and functionalized biochars (PIKE Technologies Inc.; Madison, WI, USA). Spectra were normalized the highest peak in the fingerprint region between 4000 and 550 $cm^{-1}$ to determine the functional groups on the functionalized biochars surface.

The FAMEs composition on biodiesel obtained was determined by gas chromatography coupled to mass spectrometry (GC-MS) (Agilent Technologies; Santa Clara, CA, USA), Agilent column 19091S-933E HP-1MS (30 mm × 250 μm × 0.25 μm). The carrier gas was helium (99.99%), with a flow rate of 1.1 mL min$^{-1}$. The initial temperature of the column oven kept at 140 °C for one minute, and then the temperature was increased at a rate of 8 °C min$^{-1}$ to 210 °C, then at 2 °C min$^{-1}$ up to 260 °C, in the end, heating was continued at 30 °C min$^{-1}$ up to 280 °C. The injection volume of the samples was 2 μL. Biodiesel yield was calculated by measuring the volume before and after the transesterification reaction.

## 4. Conclusions

The sulfonation process modified biochar catalysts morphology in all cases. BS-50-8 and BS-75-8 catalysts reach similar biodiesel yields than CH$_3$NaO. The highest biodiesel yield (99%) was obtained with the heterogeneous catalyst BS-25-8 at 3 wt%. It is higher than the homogeneous catalyst (93%). A better biodiesel quality was obtained using biochars than using the homogeneous catalyst. The development of CG/GI based catalysts could be integrated in the guishe biorefinery scheme previously proposed.

**Author Contributions:** Conceptualization, L.D.-J.; formal analysis, A.B.F.-D.; funding acquisition, S.C.-H.; investigation, A.B.F.-D.; supervision, S.C.-H. and L.D.-J.; writing—original draft, A.B.F.-D.; writing—review & editing, S.C.-H. and L.D.-J. All authors have read and agreed to the published version of the manuscript.

**Funding:** This research was funded by Mexican National Council for Science and Technology (CONACYT), Master Fellowship and the PEI project 222185.

**Data Availability Statement:** Data is contained within the article.

**Acknowledgments:** Authors thank Leopoldo Javier Ríos González for the support for the preparation of guishe and Enremex S. de R.L de C.V for providing the crude glycerol.

**Conflicts of Interest:** The authors declare no conflict of interest. The funders had no role in the design of the study; in the collection, analyses, or interpretation of data; in the writing of the manuscript, or in the decision to publish the results.

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
