# Peer review of "Crude Glycerol/Guishe Based Catalysts for Biodiesel Production: Conforming a Guishe Biorefinery"

_catalysts, doi:10.3390/catal11010003_

Round 1
Reviewer 1 Report
The work entitled “Crude glycerol/guishe based catalysts for biodiesel 3 production: conforming a guishe biorefinery.” reports new data regarding the preparation of sustainable acid catalysts from biomass waste founds in biorefinery. These catalysts are tested in the biodiesel production via transesterification process. The concept of using biorefinery waste to produce a heterogeneous catalyst for biorefinery reactions is of interest and constitutes one of the stronger points of this work. English level is adequate, though some effort should be made in improving the readability of the manuscript (see sentences as those in lines 81-82, line 264, or lines 273-274). The introduction is clear, and the goals of this work are aligned with the scope of Catalysts. However, some references are poorly chosen, see for instance ref #1 and #7, published in medical journals, or the lack of references to seminal works or reviews related to the topic. The authors are encouraged to improve the selection of references and to remove those published in journals devoted to areas not covered in the paper. The experimental section provides enough detail regarding the preparation, characterization, and activity tests of the catalysts, which ensures the reproducibility of the work. Some additional characterization tests are required on the catalysts. As for the results section, unfortunately the discussion shows many flaws and it strongly needs to be improved. In sum, I can not recommend the publication of this work in Catalyst at its current stage. Find below my suggestions to improve the manuscript before another submission attempt is made.
Proximal analysis: I find hard to believe that crude glycerol has a fixed carbon of 74.73%. If this were correct, TG analyses on Figure 1 should be reaching final weights well over 50% at 900 ºC, which is clearly not happening. I guess authors are grouping the volatiles fractions with fixed carbon. The authors must follow ASTM norms concerning the determination of volatiles and fixed carbon in biomass and report the full proximate analyses of the raw materials. In addition to this, the sum of moisture, ash and fixed carbon for both CG and GI does not reach 100 %, what is the missing component?
TG, Figure 1: include the DTG profiles, so that the decomposition of lignin, hemicellulose and cellulose can be clearly distinguished.
Biochar yield, Figure 2: the discussion of these results is wrong, read carefully what is said in lines 116-118. In addition, the authors should include the air TG profiles of the biochars, so that the reader can know how much inorganic matter (ash) remains in the solids.
Morphology of the catalysts. The authors should read about the reactivity of cellulose and hemicellulose with H2SO4 and discuss their results in accordance to what they find.
Porosity of the chars. Note that all the pores observed in Figure 3 are indeed macropores. However, mesopores and micropores, along with surface area, needs to be reported. The activity of the catalysts can be ruled by the porosity, and the readers could be highly interested whether these biochars have high development porosity or can be used in other applications. The authors must measure N2 & CO2 adsorption isotherms and estimate the most relevant porosity parameters. They should also report bulk and water density of the biochars and catalysts.
Elemental composition in Table 2. Some inconsistences are observed in this table. First, inorganic matter as measured by EDS needs to be compared to ash determined by air TG. Next, oxygen amount in sulfonated samples makes no sense, since sulfur is in the form of sulfates. Finally, silicon magnesium and phosphorus contents are inconsistent in sulfonated samples. Similar comments can be drawn from Table 3.
Acidity of the catalysts. The authors certainly need to include some measurement of the acidity of their catalysts, so that the reader can learn about the impact of carbonization temperature and GC ratio upon the functionalization process with sulfur acid.
Activity test. There are intriguing results here. I wonder how the authors can explain that conversion decreases with catalyst loading? In addition, since all the conversion values are so high, it is hard to understand if any differences arise from changing the GC-GI ratio or the carbonization temperature. The authors should have selected softer conditions in order to highlight the differences between catalysts.
Reusability of the catalysts. This issue is of critical relevance for heterogeneous catalysts.
Author Response
The answers are included in the attached document.

Reviewer 2 Report
The manuscript by Figueroa-Diaz et al. describes the preparation, characterization, and biodiesel production of crude glycerol and/or guishe based heterogeneous catalysts. Firstly, the elemental analysis was performed toward there two organic wastes, crude glycerol and guishe. The most significant difference between these two materials was the calcium content: guishe has much higher calcium content than crude glycerol, resulting in higher ash content. Then, pyrolysis conditions were tested to determine the optimal condition to transfer the raw material to biochar. The resulting biochar, used as the catalysts in biodiesel production, were analyzed by SEM, FT-IR, and thermogravimetric analysis. Lastly, the heterogeneous catalysts were tested in biodiesel production and a conventional homogeneous catalyst, CH3NaO. The results showed that the catalysts in this study lead to comparable or even higher yields of the product to the homogeneous catalyst. Moreover, the final products produced by heterogeneous catalysts have fewer compounds than the homogeneous catalyst, as alkyl chain breakdown is not likely to happen using heterogeneous catalysts.
This manuscript’s topic is significant and interesting: using waste from bioprocessing to prepare biofuel production catalysts. These catalysts will make biofuel production cheaper and more sustainable. The results, the product yield/quality, look great, making the application of these catalysts promising. The paper is easy to read, the introduction is clear and helpful for readers, and the results were well organized. I will recommend the acceptance after the authors address the following points.
- The author may want to define the abbreviations at first mention. These include but not limited to: "SEM, FT-IR and TGA) in Line 15, and "DTA" in Line 101.
- Line33, "CO2 emissions are reduced during combustion up to 78%". I concern about how to reduce the CO2 emissions in combustion; what is the reason? The authors refer to reference [2]. However, I didn't find the number in the paper.
- Line 36-37, the reaction with methanol or/and ethanol lead to fatty acid methyl/ethyl esters (FAME/FAEE)
- Line 40, what is "sodium hydroxide and methide or potassium"?
- Line 51, the authors may want to add details about the mechanism of using biochar as the catalyst in biodiesel production: how does the heterogeneous catalyze transesterification reactions.
- Line65, "Ixtle represents only 14.6% while the residue (guishe) 85.4% wt" is not clear.
- Line 122, the author may want to define the sample name somewhere in the paper, including B-XX-X and BS-XX-X.
- Line 173, "S=O=S" or "SO2"?
- Figure 5, right figure, "2250" wavenumber is wrongly labeled.
- Figure 5 and Line 179. Based on Table 3 and Table 2, the biochars obtained at 900oC have less S contents than the biochars abstained at 800oC. However, the SO2 peaks are more pronounced for 900oC samples than 800oC. What is the reason?
- Table 5 and 6, the author may want to add detected FAME numbers in the tables, as the author claims the biochar catalyzed reaction produced fewer FAMEs.
- Line 314, "than" to "as"
Author Response

(The authors gave the same response as above.)

Reviewer 3 Report
The described results correspond to the widely studied topic of waste management from various technologies and the production of functional materials based on them. In this manuscript, Authors analyzed the possibility of creating carbon materials from crude glycerol (obtained from biodiesel technology) and lignocellulose biomass (byproduct of ixtle production) used at different proportions. Furthermore, they tested the produced carbons as catalysts in the soybean oil transesterification reaction. The work is not groundbreaking, although it does contain some elements that could be of interest to a wider audience of Catalysts. However, it is required to answer some of the following questions.
- How to understand the composition of the raw materials, presented in Table 1, which does not give 100%? Please add a relevant comment. In this context, the determined composition should correspond with the results of TGA.
- The loss of mass during the TGA analysis (Figure 1), as well as the biochar yield (Figure 2) compared to the analysis of the composition of the raw materials suggest that the inorganic part decomposes during heating. However, this issue has not been properly discussed.
- What are the specific surface areas and pore volumes of the produced biochars?
- Do Authors assume that there are any differences in the degree of carbon graphitization in the samples calcined at 800 and 900°C? If so, does it not affect the distribution of sulfur-containing functional groups (revise the text in lines 177-179)?
- It is very interesting if the carbon catalysts can be reused after the process. Did Authors studied the performance of catalysts in successive cycles? Were any changes in the surface composition observed? The possibility of releasing sulfur compounds into the reaction products seems to be particularly important, which, from an ecological point of view, would exclude using such sulfonated carbon catalysts in the commercial transesterification processes.
Author Response

(The authors gave the same response as above.)

Round 2
Reviewer 3 Report
To be honest Authors have not fully taken into account my comments, especially those related to the issue of the release of sulfur compounds into the reaction environment, but in general I can accept the revised version of the manuscript.